# Axioms for AI Alignment from Human Feedback

**Luise Ge**
Washington University in St. Louis
g.luise@wustl.edu

**Daniel Halpern**
Harvard University
dhalpern@g.harvard.edu

**Evi Micha**
Harvard University
emicha@seas.harvard.edu

**Ariel D. Procaccia**
Harvard University
arielpro@g.harvard.edu

**Itai Shapira**
Harvard University
itaishapira@g.harvard.edu

**Yevgeniy Vorobeychik**
Washington University in St. Louis
yvorobeychik@wustl.edu

**Junlin Wu**
Washington University in St. Louis
junlin.wu@wustl.edu

## Abstract

In the context of reinforcement learning from human feedback (RLHF), the reward function is generally derived from maximum likelihood estimation of a random utility model based on pairwise comparisons made by humans. The problem of learning a reward function is one of preference aggregation that, we argue, largely falls within the scope of social choice theory. From this perspective, we can evaluate different aggregation methods via established axioms, examining whether these methods meet or fail well-known standards. We demonstrate that both the Bradley-Terry-Luce Model and its broad generalizations fail to meet basic axioms. In response, we develop novel rules for learning reward functions with strong axiomatic guarantees. A key innovation from the standpoint of social choice is that our problem has a *linear* structure, which greatly restricts the space of feasible rules and leads to a new paradigm that we call *linear social choice*.

## 1 Introduction

The alignment of AI models with human values is widely recognized as a crucial task. A prominent method for this task, *reinforcement learning with human feedback* (RLHF), has been used in different applications, such as robotics [4, 17] and recommendations [28, 1]. Recently, RLHF has attracted significant attention as a tool for fine-tuning large language models (LLMs) [22, 32, 26]. A typical implementation of RLHF involves learning a reward model using a pre-trained LLM, which is then utilized to fine-tune an existing LLM. During the learning step, human feedback is provided in the form of ordinal comparisons, and a reward function is learned from these. The most common learning method assumes an underlying *random utility model* such as the *Bradley-Terry-Luce (BTL)* model [5, 22, 8] and computes a reward function that corresponds to a maximum likelihood estimator for the observed comparisons.

Is this the "right" way of aggregating individual preferences towards a socially desirable reward function? To answer this question, we draw on *social choice theory*, a field that studies collective decision making through a mathematical lens [6]. The maximum likelihood estimation approach is in line with a well-established body of work that assumes that different human participants have preferences stemming from noisy estimation of a common ground truth, and the goal is to learn this ground truth as accurately as possible [29]. But this is not the case when it comes to questions of AI

alignment, where individuals can have legitimate differences of opinion rooted in different values or priorities.

We argue that when preferences are truly heterogeneous, the *axiomatic approach*—which rose to prominence in social choice with the work of Arrow [2]—may be more suitable. This approach analyzes the desirability of aggregation methods by their satisfaction of certain axioms that capture notions of consensus, fairness, and economic efficiency. Specifically, we are interested in the axiomatic properties of aggregation methods that take ordinal preferences as input and output a reward function. We address the following two research questions: *What axioms are satisfied by aggregation methods used by existing RLHF algorithms? And are there alternative aggregation methods that offer stronger axiomatic guarantees?*

## 1.1 Our Approach

In social choice theory, axioms are typically defined for rules that map rankings over candidates to a single winner (social choice functions) or a ranking of the candidates (social welfare functions). By contrast, we are interested in rules that assign a reward to each candidate. This gap is easy to bridge, though: we simply consider a ranking of the candidates by in descending reward order.

A much more significant gap is that in classical social choice, all relevant candidates appear in the input preferences, whereas in our setting (where candidates correspond, e.g., to prompts and their responses), we are only given preferences over a relatively small set of candidates *identified by their (known) features*, and we need to generalize from this information. In practice, this entails using a restricted—commonly, parametric—class of reward models which map candidate features to real-valued rewards, and which we fit to existing data.

Specifically, we assume that a *linear* reward function defined by a parameter vector determines the reward of each candidate by computing the inner product of the parameter vector and the feature vector of the candidate; these modeling choices are consistent with prior and concurrent work [31, 30, 15] and aim to capture the practice of RLHF.[1] Each human participant (henceforth referred to as a *voter*) is associated with a parameter vector, which is unknown to us and is used to specify ordinal preferences over the candidates. Our task is to design *linear rank aggregation rules*, which aggregate rankings induced by these individual linear functions[2] into a collective ranking that is also induced by a linear function; this is a new paradigm in social choice, for which we coin the term *linear social choice*.

To evaluate linear rank aggregation rules, we adapt fundamental axioms from social choice theory [6]. The first is *Pareto optimality (PO)*, which requires that if a candidate $a$ is ranked above candidate $b$ in *every* input ranking, then the resulting ranking should rank $a$ above $b$. This is seen as a basic requirement and is satisfied by *every* standard voting method in the classical setting.

The second axiom is *pairwise majority consistency (PMC)*: If there exists a reward function that generates a ranking where, for each pair of candidates, a majority of voters agree with the ranking, then the resulting ranking should match that ranking. This axiom is an extension of *Condorcet consistency* to rankings, and is satisfied by some, but not all, standard voting methods in the classical setting.

## 1.2 Our Results

We start by examining, in Section 3.1, a family of loss-based rules that finds a ranking induced by a parameter vector that optimizes a measure of loss; this measure increases for every disagreement with a voter on a pair of alternatives, where the larger the difference in rewards, the larger the penalty. Crucially, by plugging in binary cross-entropy loss we can recover the BTL model. Our first main result is that whenever the loss function is weakly convex and nondecreasing, or strictly convex— conditions satisfied by binary cross-entropy loss, as well as, e.g., exponential and hinge loss—the

---

[1]We can represent the reward model as an embedding layer $\phi(x)$ which is then aggregated linearly to compute the final reward. If we fix the embedding function $\phi$ and treat its output as the feature representation of the outcomes, such as prompt-response pairs, the resulting reward model is linear in $\phi(x)$; see, e.g. [31].

[2]In practice, typical RLHF datasets consist of pairwise comparisons, not complete rankings. Assuming rankings as input makes our exposition cleaner, and it is not a fundamental limitation, as we discuss in Section 5.

corresponding rule fails both PMC and PO. This result suggests that the prevailing practice of RLHF is flawed from an axiomatic viewpoint.

In Section 3.2, we take a first step towards addressing this shortcoming. We modify the loss-based formulation to focus on majority preferences rather than individual preferences. This modification defines a family of rules that are PMC, but we show that all of them fail PO by establishing an even stronger impossibility result: In stark contrast to the classical setting, any *linear* rank aggregation rule that depends only on majority preferences must fail PO.

In order to achieve both PO and PMC, we design (in Section 4) a linear rank aggregation rule that we call *Leximax Copeland subject to PO*. Not only does it satisfy our two main axioms, it also satisfies two additional ones, *majority consistency* and *winner monotonicity*.

To summarize, while widely applied rules fail to meet basic axioms, there are alternative methods that are desirable from this viewpoint. Our approach, therefore, provides a discriminative lens through which to evaluate RLHF methods and AI alignment methods more broadly.

### 1.3 Related Work

During the eight months in which we have actively worked on this project (from September 2023 until May 2024) — and especially in the first few months of 2024 — a slew of independent, concurrent papers seeking to build bridges between social choice and RLHF have become publicly available [10, 12, 19, 30, 23, 7, 15, 27, 25]; this surge of interest points, in our view, to the importance of the agenda.

Three of those papers are position papers that conceptually support our work in that they discuss the possibility of applying an axiomatic approach to RLHF [10, 12, 19], although they do not provide any technical results. By contrast, existing technical papers on RLHF do not take an axiomatic approach. Of the concurrent technical papers, the one that is most closely related to ours is that of Siththaranjan et al. [25]. They show, among other results, that the ranking induced by the reward function that the MLE estimator of the Bradley-Terry-Luce Model returns follows the famous Borda count rule when unrestricted reward functions are allowed. In the classical setting, Borda count has strong axiomatic guarantees, including PO (but not PMC). However, it cannot be realized as a linear rank aggregation rule, and it is arguably impractical for RLHF.

Our work builds on an earlier study by Noothigattu et al. [21], which explores the axiomatic properties of reward functions defined as MLE estimators of underlying random utility models. The key difference is that their approach allows for general reward functions, not just linear ones, and they do not consider features at all. Unlike our findings, they show that the BTL model satisfies Pareto Optimality under these conditions. Additionally, they find that pairwise majority consistency is violated even without assuming linearity. However, their results strongly depend on varying the number of comparisons across different pairs of candidates. By contrast, our findings demonstrate that pairwise majority consistency is violated even when the number of comparisons is equal across all pairs of candidates.

## 2   The Linear Social Choice Model

Let $C$ be a set of $m$ distinct prompt/responses, referred to as *candidates*, and let $V = \{1, \ldots, n\}$ be a set of $n$ human participants, known as *voters*. We denote by $\mathbb{R}^d$ the $d$-dimensional real space in which both candidate feature vectors and chosen parameter vectors lie.

Each candidate $c \in C$ is associated with a distinct feature vector $\mathbf{x}_c \in \mathbb{R}^d$. A parameter vector $\theta \in \mathbb{R}^d$ induces a linear reward function $r_\theta : C \mapsto R$ defined by taking the dot product with feature vectors $r_\theta(c) = \langle \theta, \mathbf{x}_c \rangle$. We will primarily be interested in how these parameterized functions rank the candidates by reward. Let $R^{a \succ b} = \{\theta \mid r_\theta(a) \geq r_\theta(b)\}$ be the region where the reward of $a$ is at least as large as that of $b$. Note that $R^{a \succ b}$ and $R^{b \succ a}$ split $\mathbb{R}^d$ into two half spaces, separated by the hyperplane orthogonal to $\mathbf{x}_a - \mathbf{x}_b$. Parameter vectors $\theta$ on the hyperplane have $r_\theta(a) = r_\theta(b)$, while rankings in the interior of either half-space strictly rank one over the other.

For a ranking $\sigma$ over the candidates, we say that $\theta$ induces $\sigma$, denoted $\theta \rhd \sigma$, if $a \succ_\sigma b$ implies $r_\theta(a) \geq r_\theta(b)$. Let $R^\sigma = \{\theta \mid \theta \rhd \sigma\}$ be the set of vectors $\theta$ that induce it. Note that this can be written as the intersection of corresponding half spaces $R^\sigma = \bigcap_{a,b : a \succ_\sigma b} R^{a \succ b}$. Further, the

collection of $\{R^\sigma\}$ essentially form a partition of $\mathbb{R}^d$, covering the space and intersecting only at their boundaries.

We call a $\theta$ *non-degenerate* if it is fully on one side of each of the separating hyperplanes, i.e., $r_\theta(a) \neq r_\theta(b)$ for all $a, b \in C$. Non-degenerate parameter vectors lie in the interior of some $R^\sigma$, and thus induce exactly one ranking. We call $\sigma$ *feasible* if $R^\sigma$ has a nonempty interior, i.e., is induced by some nondegenerate $\theta$.[3]

Each voter $i \in V$ submits a ranking over the candidate $\sigma_i$. We assume that the feature space is rich enough that voter preferences can be captured via non-degenerate parameter vectors. In other words, we assume that each $\sigma_i$ is feasible. We refer to the vector of voter rankings $\pi = (\sigma_i)_{i \in V}$ as a *profile*. Further, for two candidates $a, b$, we write $n_{a \succ b}(\pi) := |\{i \in V \mid a \succ_{\sigma_i} b\}|$ for the number of voters that prefer $a$ to $b$, and $w_{a \succ b}(\pi) = n_{a \succ b}(\pi)/n$ for the proportion of such voters. When the profile $\pi$ is clear from context, we may shorten these to $n_{a \succ b}$ and $w_{a \succ b}$, respectively.

We define a *parameter aggregation rule* as a function that takes as input a profile $\pi$ and outputs a parameter vector $\theta^*$. Our goal is to design parameter aggregation rules such that $r_{\theta^*}$ satisfies desirable properties with respect to the voter preferences. However, as the properties we care about will only be with respect to how $r_{\theta^*}$ *ranks* the candidates, it will be more convenient to work with what we call *linear rank aggregation rules* that take as input a profile $\pi$ and output a *feasible* ranking $\sigma$. There is a natural way to interpret a parameter aggregation rule as a linear rank aggregation rule, namely, output any feasible ranking induced by $\theta^*$. The exact properties of the parameter aggregation rule could in principle be sensitive to the tie-breaking of non-degenerate outputs, however, all of our results will be robust to such tie-breaking.[4]

We pay special attention to a prominent family of rules from social choice theory referred to as $C1$ *rules* [14], whose outputs depend only on majority relationships, i.e., they only need to know for each pair of candidates $(a, b)$ whether the majority prefers $a$ or $b$.

In our study, we examine several axioms borrowed from social choice theory to evaluate the reasonableness (fairness) of our aggregation mechanisms. These axioms include:

**Definition 2.1** (Pareto Optimality). *A linear rank aggregation rule $f$ satisfies* Pareto optimality *if, whenever every voter prefers candidate $a$ over candidate $b$ on $\pi$, i.e., $w_{a \succ b}(\pi) = 1$, then candidate $a$ is ranked higher than candidate $b$ in the output ranking, i.e., $a \succ_{f(\pi)} b$.*

**Definition 2.2** (Pairwise Majority Consistency (PMC)). *A ranking $\sigma$ is called a* PMC *ranking for profile $\pi$ if for all $a, b \in C$, $a \succ_\sigma b$ if and only if a majority of voters rank $a \succ_{\sigma_i} b$, i.e., $w_{a \succ b} > 1/2$. A linear rank aggregation rule satisfies PMC if, when a PMC ranking $\sigma$ exists for the input profile $\pi$ and $\sigma$ is feasible, then $f(\pi) = \sigma$.*

Note that a PMC ranking for each $\pi$ need not exist, but when one does, it is unique. The words "$\sigma$ is feasible" allude to the possibility that no non-degenerate parameter vector $\theta$ induces the unique PMC ranking. Indeed, we have such an example; see Appendix B for details.

Our research question, then, is whether these axioms can be simultaneously satisfied by *linear* rank aggregation rules. Our approach seeks to provide a concrete illustration of how theoretical insights from social choice can inform practical algorithm design in RLHF.

---

[3] The number of feasible rankings is in general upper bounded by $m^{O(d)}$ due to how many regions $\binom{m}{2}$ hyperplanes can partition the space into [13]. Further, under mild conditions on the feature vector locations, the exact number of feasible rankings is known [11, 16].

[4] One may wonder if there are any computational barriers to converting between parameter and linear rank aggregation rules. However, this is not the case. In particular, for *every* set of pairwise comparisons $R = \{a_1 \succ b_1, a_2 \succ b_2, a_3 \succ a_3, \ldots\}$, we can efficiently (i) check if there is a feasible ranking $\sigma$ consistent with $R$, and (ii) if such a $\sigma$ exists, find a nondegenerate $\theta$ inducing such a $\sigma$. This can be done by finding a $\theta$ satisfying the following system of linear inequalities, or determining that no such $\theta$ can satisfy them (which can be done using a linear program): $r_\theta(a) \geq r_\theta(b) + 1$, $\forall a \succ b \in R$. Any $\theta$ satisfying the system would be nondegenerate and induce a $\sigma$ consistent with $R$. Furthermore, if a feasible ranking $\sigma$ is consistent with $R$, then taking any nondegenerate $\theta$ inducing $\sigma$ and scaling up its values would satisfy all inequalities. This means that given a ranking $\sigma = c_1 \succ c_2 \succ \cdots \succ c_m$, we can check whether or not it is feasible, and so, find a $\theta$ inducing it by running this with $R = \{c_1 \succ c_2, \ldots, c_{m-1} \succ c_m\}$. Additionally, given a possibly degenerate $\theta$, we can find a feasible ranking $\sigma$ which $\theta$ induces by running this with $R = \{a \succ b \mid r_\theta(a) > r_\theta(b)\}$.

# 3 Loss-Based Rules

## 3.1 Standard Loss Formulation

We begin our study of linear social choice by considering a quite broad yet natural class of rules that capture how RLHF is currently being done. Their core idea is the following: when considering parameter vector $\theta$, for each voter $i$ that ranks a pair of candidates $a \succ_i b$, we should incur some loss for giving $b$ a higher reward than $a$. To formalize this, let $\ell : \mathbb{R} \to \mathbb{R}$ be a *loss function*, which we assume is nonnegative. We can then choose a parameter vector minimizing

$$\mathcal{L}(\theta; \pi, \ell) = \sum_{a \neq b \in C} n_{a \succ b}(\pi) \cdot \ell(r_\theta(b) - r_\theta(a)).$$

Note that the BTL model fits within this framework using $\ell(x) = \ln(1 + e^x)$, i.e., *binary cross-entropy loss*.[5] One caveat to this approach, however, is that an optimal $\theta$ need not be well-defined: it is possible that no minimum is attained. Fortunately, since we only care about rankings induced by optimal parameter vectors, we can conveniently remedy this by saying the output is any ranking that is induced by parameter vectors that are arbitrarily close to optimal. More formally, we say that a linear rank aggregation rule $f$ *minimizes* $\ell$ if for all $\sigma = f(\pi)$,

$$\inf_{\theta:\theta \in R^\sigma} \mathcal{L}(\theta; \pi, \ell) = \inf_{\theta} \mathcal{L}(\theta; \pi, \ell). \tag{1}$$

Even if no minimum is attained, there is always a choice of feasible ranking $\sigma$ such that Equation (1) is satisfied.

With this definition in hand, we proceed to our first main result, which spells rather bad news for this class of rules: *Any* loss-based aggregation rule using a nondecreasing and convex loss function (of which BTL is one, and hinge loss is another) will fail our two core axioms, PMC and PO. This paints a negative picture for current RLHF methods with respect to their social choice guarantees. Note that we will exclude the discussion of loss functions with a global minimum at zero, like ReLU, because the loss minimizer will be zero, making all rankings vacuous consequently. And we have focused on convex loss functions due to their practical optimization ease.

**Theorem 3.1.** *If a linear rank aggregation rule $f$ optimizes a loss function $\ell$ that satisfies $\inf_x \ell(x) < \ell(0)$ and is either nondecreasing and weakly convex, or strictly convex (and possibly nonmonotone), then $f$ fails PMC and PO.*

*Proof.* Fix a loss function $\ell$ satisfying the theorem conditions. Note that since $\ell$ is convex, we may also assume it is continuous [24, Corollary 10.1.1]. Furthermore, since $\inf_x \ell(x) < \ell(0)$, we know that there exists $x \neq 0$ such that $\ell(x) < \ell(0)$. The case where $x > 0$ is relatively simple (as such loss functions lead to unnatural behavior), and we handle it at the end of the proof. For now, we assume that there exists $x < 0$ such that $\ell(x) < \ell(0)$. Note that this also implies that for all $y \geq 0$, $\ell$ is lower bounded by the affine linear function connecting $(x, \ell(x))$ and $(0, \ell(0))$, and thus, $\lim_{x \to \infty} \ell(x) = \infty$.

We begin with a small instance of just three candidates $C^{core} = \{a, b, c\}$ to gain some traction on how $\ell$ behaves. We will later extend this instance with additional candidates to demonstrate a profile where PO and PMC fail. The candidates will have feature vectors $\mathbf{x}_a := (2, 1)$, $\mathbf{x}_b := (1, 1)$, and $\mathbf{x}_c := (0, 0)$, respectively. Furthermore, a $p$-fraction of voters (for $p$ to be chosen later) will rank $a \succ b \succ c$, while the remaining $(1 - p)$-fraction will have inverted preferences, ranking $c \succ b \succ a$.[6]

Let

$$\mathcal{L}^{core}(\theta) := \sum_{x \neq y \in C^{core}} w_{x \succ y} \ell(r_\theta(y) - r_\theta(x))$$

---

[5]Typically, BTL is presented as choosing $\theta$ maximizing the likelihood of seeing the pairwise comparisons we observed, assuming that $\Pr[a \succ b] = \frac{e^{r_\theta(a)}}{e^{r_\theta(a)} + e^{r_\theta(b)}}$. That is, we choose $\theta$ maximizing $\prod_{a \neq b} \left( \frac{e^{r_\theta(a)}}{e^{r_\theta(a)} + e^{r_\theta(b)}} \right)^{n_{a \succ b}}$. By taking the log and swapping the sign, we see that this is equivalent to minimizing $\sum_{a \neq b} \log \left( 1 + e^{r_\theta(b) - r_\theta(a)} \right)$.

[6]It so happens that these rankings are feasible, but for now we will not worry about this as loss function-based rules still make sense regardless of whether the inputs are feasible. For the final example, we will ensure that the rankings are feasible.

with $w_{x \succ y} \in \{1 - p, p\}$ be the loss function on this instance (scaling $n_{x \succ y}$ down to $w_{x \succ y}$ leads to an equivalent formulation). Let $g(x) = p \cdot \ell(-x) + (1-p)\ell(x)$. Note that we can rewrite $\mathcal{L}^{com}$ as

$$\mathcal{L}^{core}(\theta) = g(r_\theta(a) - r_\theta(b)) + g(r_\theta(a) - r_\theta(c)) + g(r_\theta(b) - r_\theta(c)).$$

Note that $r_\theta(c) = 0$ for all $\theta$, so we can simplify this to

$$\mathcal{L}^{core}(\theta) = g(r_\theta(a) - r_\theta(b)) + g(r_\theta(a)) + g(r_\theta(b)).$$

We will consider an unconstrained version of this problem where we are free to choose rewards $r_a, r_b \in \mathbb{R}$ arbitrarily, and later show by which vectors $\theta$ these optimal values can be induced. That is, we will first find $r_a, r_b \in \mathbb{R}$ minimizing

$$\mathcal{L}^{unconstr}(r_a, r_b) := g(r_a - r_b) + g(r_a) + g(r_b).$$

Let $OPT^{core} = \{\theta \mid \mathcal{L}^{core}(\theta) = \inf_{\theta'} \mathcal{L}^{core}(\theta')\}$ and $OPT^{unconstr}\{(r_a, r_b) \mid \mathcal{L}^{unconstr}(r_a, r_b) = \inf_{r'_a, r'_b} \mathcal{L}^{unconstr}(r'_a, r'_b)\}$ be the set of minimizers for these two loss functions. In Appendix A, we establish the following results about these optimal sets.

**Lemma 3.2.** *There exists a rational* $p \in (1/2, 1]$ *and values* $A_1 < A_2$ *with* $A_2 > 0$ *such that* $OPT^{unconstr}$ *is nonempty and for all* $(r_a, r_b) \in OPT^{unconstr}$, $r_a > A_2$ *and* $r_b \leq A_1$.

**Lemma 3.3.** *Suppose Lemma 3.2 holds for values* $p, A_1$ *and* $A_2$, *then, for this same choice of* $p$, $OPT^{core}$ *is nonempty and there exist* $A_3$ *and* $A_4$ *with* $A_3 > 0$ *such that for all* $(\theta_1, \theta_2) \in OPT^{core}$, $\theta_1 > A_3$ *and* $\theta_2 < A_4$.

We will now explicitly construct a family of instances with candidate feature vectors parameterized by a value $\varepsilon \in \mathbb{R}$ such that for sufficiently small $\varepsilon > 0$, the output of $f$ fails the two axioms. Fix $p, A_3$ and $A_4$ from Lemma 3.3, and choose $\delta$ with $0 < \delta < 1$ such that $\delta A_4 - A_3 < 0$ ($\delta < A_3/A_4$ works if $A_4 > 0$, and otherwise, any $0 < \delta < 1$ will do).

Each instance will have six candidates, which we will think of as two groups of three, $C = C^{core} \cup C^{copies}$. The first group $C^{core} = \{a, b, c\}$ will be the same as the three-candidate instance from above, while the second group $C^{copies} = \{a', b', c'\}$ will be new. The candidates $a, b, c$ will still be located at $\mathbf{x}_a := (2, 1)$, $\mathbf{x}_b := (1, 1)$, and $\mathbf{x}_c := (0, 0)$, respectively. The candidates $a', b', c'$ will be located near their undecorated counterparts at $\mathbf{x}_{a'} := \mathbf{x}_a + (-\varepsilon, 0)$, $\mathbf{x}_{b'} := \mathbf{x}_b + (-\varepsilon, 0)$ and $\mathbf{x}_{c'} := \mathbf{x}_c + (-\varepsilon, \delta \cdot \varepsilon)$.

Next, we describe the voter preferences. A $p$-fraction of voters will have the ranking $a \succ a' \succ b \succ b' \succ c' \succ c$, and the remaining $(1-p)$-fraction of voters will have ranking $c' \succ c \succ b' \succ b \succ a' \succ a$. As long as $0 < \varepsilon < 1$ (which will be the case for our final chosen $\varepsilon$), these are both feasible rankings. The former is induced by the nondegenerate feature vector $(1, 1)$[7] and the latter by $(-1, 0)$.[8]

For each $\varepsilon \in \mathbb{R}$, let

$$\mathcal{L}^\varepsilon(\theta) = \sum_{x \neq y \in C} w_{x \succ y} \ell(r_\theta(y) - r_\theta(x))$$

with $w_{x \succ y} \in \{0, 1 - p, p, 1\}$ be the loss function we are optimizing using candidate locations parameterized by $\varepsilon$.

We will show that for sufficiently small $\varepsilon > 0$, $\inf_{\theta \in R^{c' \succ c}} \mathcal{L}^\varepsilon(\theta) > \inf_\theta \mathcal{L}^\varepsilon(\theta)$. This means that $f$ must output a ranking with $c \succ c'$. Observe that this is a PO violation because all voters agree that $c' \succ c$. Furthermore, this is a PMC violation because a majority of voters have the ranking $a \succ a' \succ b \succ b' \succ c' \succ c$, yet this is not the output.

Let $OPT(\varepsilon)$ be the set of vectors optimizing $\mathcal{L}^\varepsilon$. The rest of the proof will follow from the following two lemmas, whose proofs are in Appendix A.

**Lemma 3.4.** $OPT(0) \subseteq \overline{R^{c' \succ c}}$.

**Lemma 3.5.** *Suppose* $OPT(0) \subseteq \overline{R^{c' \succ c}}$, *then, for sufficiently small* $\varepsilon > 0$, $\inf_{\theta: \theta \in R^{c' \succ c}} \mathcal{L}^\varepsilon(\theta) > \inf_\theta \mathcal{L}^\varepsilon(\theta)$.

---

[7]This induces rewards $3, 3 - \varepsilon, 2, 2 - \varepsilon, (1 - \delta) \cdot \varepsilon$ and $0$ for $a, a', b, b', c'$, and $c$, respectively.
[8]This induces rewards $\varepsilon, 0, -1 + \varepsilon, -1, -2 + \varepsilon$ and $-2$. for $c', c, b', b, a'$, and $a$, respectively.

Finally, we handle the case that exists $x > 0$ such that $\ell(x) < \ell(0)$. Note that by convexity, this implies that for all $y < 0$, $\ell(y) > \ell(0) > \ell(x)$, so $\inf_{y \leq 0} \ell(y) < \inf_y \ell(y)$. Now, consider an instance with two candidates $\{a, b\}$ located at $\mathbf{x}_a = (1, 0)$ and $\mathbf{x}_b = (0, 1)$, and a single voter ranking $a \succ b$ (feasible via the parameter vector $(1, 0)$). It is possible to achieve a loss of $\ell(x)$, e.g., by outputting the parameter vector $(0, x)$. On the other hand, any $\theta$ inducing the ranking $a \succ b$ will be lower bounded by $\ell(0) > \ell(x)$ from above. Hence, $f$ must output $b \succ a$, which is both a PO and PMC violation. $\qquad\square$

### 3.2 Majority-Based Loss Formulation

Despite the negative results for loss-function-based rules, we may hope for a remedy using slightly different information. Specifically, we consider a similar loss-based function that rather than getting penalized for disagreeing with each voter only gets penalized if it disagrees with a majority of voters. That is, we choose $\theta$ minimizing

$$\mathcal{L}^{maj}(\theta; \pi, \ell) = \sum_{a \neq b \in C} \mathbb{I}[w_{a \succ b}(\pi) > 1/2] \cdot \ell(r_{\theta(b)} - r_{\theta(a)}).$$

Defining a parameter aggregation function based on this loss suffers from the same caveat as before, that in some cases no optimal $\theta$ exists. Nevertheless, we can apply an analogous fix for a ranking variant. We say that a linear rank aggregation rule $f$ *minimizes $\ell$ in the majority formulation* if for all $\sigma = f(\pi)$,

$$\inf_{\theta: \theta \in R^\sigma} \mathcal{L}^{maj}(\theta; \pi, \ell) = \inf_\theta \mathcal{L}^{maj}(\theta; \pi, \ell).$$

We first show (in Appendix A.5) that this does indeed help achieve PMC with essentially all loss functions.

**Theorem 3.6.** *Fix a nondecreasing loss function $\ell$ with $\ell(0) > \inf_x \ell(x)$. If a linear rank aggregation rule $f$ minimizes $\ell$ in the majority formulation, then $f$ satisfies PMC.*

Note that if the $\ell(0) > \inf_x \ell(x)$ condition is not satisfied, i.e., $\ell(0) = \inf_x \ell(x)$, then *all* linear rank aggregation rules $f$ minimize $\ell$ in the majority formulation, so satisfying this is a vacuous condition. Indeed, the parameter vector $\mathbf{0}$ of all 0s achieves optimal loss of $\ell(0)$ for each pair and is consistent with every ranking $\sigma$. Therefore, the condition $\ell(0) > \inf_x \ell(x)$ is as innocuous as possible to rule out these edge cases.

However, despite this good news for PMC, we show that this does not help in achieving PO. In fact, our negative result extends to every $C1$ linear rank aggregation rule. Note that if $f$ minimizing $\ell$ in the majority formulation breaks ties consistently (i.e., if multiple feasible rankings are optimal, then it consistently chooses the same one), then it is C1. We then have the following result, whose proof is relegated to Appendix A.6.

**Theorem 3.7.** *All $C1$ linear rank aggregation rules fail PO.*

This result is quite unfortunate, because if there were a rule that is both $C1$ and PO, we would automatically achieve PMC: Whenever there is a feasible PMC ranking, a $C1$ rule cannot distinguish between this profile and a profile where all voters submit this ranking, hence, under the PO criterion, it must output it. Furthermore, whenever there is a PMC ranking, outputting it is necessarily PO, as for every pair, a majority of voters agree with the PMC ranking. Interestingly, in the proof, we construct a profile which has a PMC ranking, yet it is not feasible, and no matter how a $C1$ linear rank aggregation rule breaks ties, there is an underlying profile in which this output violates PO.

## 4  Social Choice-Based Rule

In light of the above negative results, in this section, we ask whether there are linear rank aggregation rules that concurrently satisfy our two core axioms, PO and PMC. We answer this question affirmatively by presenting a new method based on a prominent rule from voting theory.

The *Copeland rule* assigns a *Copeland score* to each alternative equal to the number of other alternatives it beats in a pairwise competition, i.e., the score for $a$ is $|\{b \mid w_{a \succ b} > 1/2\}|$. It then ranks the candidates in descending order according to their Copeland scores (breaking ties arbitrarily). It

is known that Copeland satisfies PO, PMC, and additional axiomatic properties. However, in linear social choice, since not every ranking is feasible, we cannot always output the Copeland ranking.

We, therefore, define a new linear rank aggregation rule, which we call *leximax Copeland*. This rule chooses a feasible ranking as follows. It ranks first the candidate with the highest Copeland score that can be feasibly ranked first under some parameter vector $\theta$. Subject to this first position, it ranks second the candidate with the highest Copeland score which can be feasibly ranked second, and continues this process for subsequent positions.

Copeland's rule is a $C1$ rule because it only requires the majority relationships between the candidates. Analogously, leximax Copeland is also a $C1$ linear rank aggregation rule. Therefore, by Theorem 3.7, it does not satisfy the PO criterion. To address this issue, we define a variant called *leximax Copeland subject to PO (LCPO)*, which incorporates the PO criterion. Under LCPO, for every pair of alternatives where one dominates the other, the rule restricts rankings to place the dominating alternative above the dominated one.

The rule remains well-defined since the set of feasible rankings when enforcing the PO criterion is non-empty, as whenever $a$ dominates $b$, all the rankings in the input profile rank $a$ above $b$. Note that if the Copeland ranking is feasible, then this rule outputs that ranking, since unrestricted Copeland satisfies PO.

In addition to PO and PMC, we wish to show that LCPO satisfies two additional properties, which we define presently.

**Definition 4.1** (majority consistency). *A linear rank aggregation rule satisfies* majority consistency *if when a candidate $a$ is ranked first by a majority of voters in the input profile, $a$ is ranked first in the output ranking.*

Majority consistency ensures that the collective decision reflects the preference of the majority when there is a clear favorite. This principle aligns with PMC, but specifically focuses on the majority's favorite alternative. However, as we discussed above, a PMC ranking does not necessarily exist, and even when it exists, it is not necessarily feasible. By contrast, when a majority winner exists, this candidate is necessarily ranked first by a majority of voters in the input profile, who themselves (by assumption) submit feasible rankings. Therefore, we need not handle the case where it is impossible to rank the majority winner first.

**Definition 4.2** (winner monotonicity). *A linear rank aggregation rule satisfies* winner monotonicity *if, when a candidate $a$ is ranked first in the output ranking, elevating $a$ in any voter's preference does not cause $a$ to lose their top position in the updated aggregate ranking.*

Winner monotonicity ensures that improving a leading candidate's position among individual voters will not result in that candidate's demotion.

We now state and prove the main result of this section.

**Theorem 4.3.** *LCPO satisfies PO, PMC, majority consistency and winner monotonicity.*

*Proof.* LCPO trivially satisfies PO since it always outputs a ranking that respects the PO criterion. Moreover, since Copeland satisfies PMC, and whenever Copeland's ranking is in the domain, leximax Copeland subject to PO returns this ranking, it clearly satisfies PMC.

Note that if an alternative $a$ is ranked first by at least half of the voters, then $a$ has the highest Copeland score, meaning that leximax Copeland subject to PO will rank this candidate first if this is possible. We see that this is indeed possible, by noticing that there is at least one feasible ranking in the input profile where $a$ is ranked first, and any such input ranking is feasible (by assumption) and satisfies the PO requirement. Therefore, majority consistency is satisfied.

It remains to show that LCPO satisfies winner monotonicity. Suppose that on input profile $\pi$, the rule outputs a ranking $\sigma$ where candidate $a$ is ranked first. Now, consider a profile $\pi'$ which is similar to $\pi$ with the only exception being a ranking in which $a$ is placed in a higher position. Let $S$ be the set of agents that ranked above $a$ in Copeland's ranking under $\pi$ and let $S'$ be the set of agents that ranked above $a$ in Copeland's ranking under $\pi'$. Note that $S' \subseteq S$, since when moving from $\pi$ to $\pi'$, only the Copeland score of $a$ can increase, and therefore it is not possible for a candidate $b$ to beat $a$ under $\pi'$ but not under $\pi$.

Now, suppose that $R$ and $R'$ are the set of rankings that satisfy the PO criterion with respect to $\pi$ and $\pi'$, respectively. We show that $R' \subseteq R$. First, note that since $a$ is ranked first under $\pi$, no alternative dominates $a$ in $\pi$, as otherwise the PO criterion would be violated. Therefore, we get that no other alternative dominates $a$ in $\pi'$ as well. Moreover, note that if $b$ dominates $c$ in $\pi$, then this remains true in $\pi'$ as well. On the other hand, it is possible that $a$ dominates an alternative $b$ in $\pi'$ but not in $\pi$. From all of the above, we conclude that $R' \subseteq R$.

Since $a$ is ranked first in $\sigma$, we get that for every candidate $b \in S$, there is no ranking in $R$ in which $b$ is ranked first, since otherwise, LCPO would output such a ranking. This also means that for every candidate $b$ in $S'$, there is no ranking in $R'$ in which $b$ is ranked first, since $R' \subseteq R$ and $S' \subseteq S$. Moreover, note that every ranking in $R$ in which $a$ is ranked first is also in $R'$ since it satisfies all the PO restrictions of $\pi'$. Therefore, under $\pi'$, LCPO outputs a ranking in which $a$ is ranked first. □

Leximax Copeland subject to PO can be implemented in polynomial time by solving $O(|C|^2)$ relatively small linear programs. Specifically, given an input profile, we sequentially choose the candidate that is ranked in position $r + 1$ as follows. We denote by $\sigma_r$ the partial ranking, where the first $r$ positions have been fixed. For each candidate $c$ that has not been ranked yet, we want to check if there is a parameter vector that adheres to the partial ranking $\sigma_r$, respects the Pareto optimality criterion and ranks $c$ at position $r + 1$. Since all these constraints can be expressed as pairwise comparisons, we can use a linear program such as the one described in Footnote 4 to check if such a feasible ranking exists. Among the candidates meeting this criterion, we select the one with the highest Copeland score for position $r$.

## 5   Discussion

We conclude with a discussion of several extensions and limitations of our approach and results.

First of all, we wish to emphasize that our results are theoretical. While they highlight some shortcomings of the current practice of RLHF, our goal was not to "outperform" existing RLHF methods. Rather, we see our model as giving a framework for understanding and comparing rules and methods — it is a (useful, we believe) lens through which researchers and engineers can examine their AI alignment methods.

Second, as written, our model has voters give their complete rankings, while in practice, this would be infeasible. In the real world, we are likely to elicit only relatively few pairwise comparisons per person. For our negative results, this assumption only makes them stronger: the BTL model fails both PO and PMC *even* with access to complete voter rankings. By contrast, for the positive results, specifically implementing leximax Copeland subject to PO, this ostensibly seems like a serious limitation. However, the complete rankings are not necessary for computing this rule, rather, all we need to know are PO dominance relationships and majority directions. We can therefore apply the rule whenever we can approximate this information, for example, through sampling. An alternative approach is to infer a complete ranking of each voter by fitting a parameter vector based on their pairwise responses; this process of learning a complete ranking and then running voting rules has been used before in a variety of settings [20, 18].

Third, our work initiates the study of the axiomatic method in our linear social choice model. However, we leave open many questions about which axioms are compatible and finding rules that achieve them. It should be clear by now that the primary challenge in linear social choice is that not every ranking over the candidates can be output. This means that essentially all known aggregation rules cannot directly be used without at least some modification. A natural direction to tackle is to try to find methods of converting known voting rules into linear aggregation ones while maintaining some of their axiomatic properties. To this end, we conclude with some preliminary results, and somewhat surprising findings within this space.

Some rules which optimize over rankings can be naturally transformed. For example, consider the Kemeny rule, which returns the ranking with the smallest pairwise disagreement over all votes. This can easily be transformed to the linear setting by simply outputting the optimal feasible ranking. In fact, in Appendix C.2, we show that this rule carries over the property of *separability*,[9] a social choice axiom that is violated by Copeland (in the classical setting) and leximax Copeland subject to

---

[9]Formally, *ranking separability* to distinguish it from the single-winner version.

PO (in our setting). We show this in Appendix C.1. However, quite strikingly, although separability remains, this transformation makes Kemeny no longer PO (Appendix C.2).

Finally, note that the "leximax" portion of leximax Copeland can be seen as a general purpose tool for mapping traditional rules to linear aggregation rules. In Appendix C.3, we explore leximax plurality (run leximax on the ranking of candidates by plurality scores), and show that it satisfies majority consistency, winner monotonicity, and separability. Additionally, the "subject to PO" can be seen as another "tool" for enforcing the Pareto optimality criterion when a rule does not independently satisfy it. However, enforcing PO can again cause somewhat surprising results. For example, in Appendix C.2, we show that linear Kemeny subject to PO, while now trivially satisfying PO, again violates separability. These observations indicate the challenges inherent in linear social choice, and we hope these open questions inspire fruitful follow-up research.

## Acknowledgments

This research was partially supported by the National Science Foundation under grants IIS-2147187, IIS-2229881, CCF-2007080, IIS-1905558, and IIS-2214141; and by the Office of Naval Research under grants N00014-20-1-2488 and N00014-24-1-2704.

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

# A  Deferred Proofs

## A.1  Proof of Lemma 3.2

*Proof.* We begin with some observations on $g$. First, we have that since $\ell$ is nonnegative, $g$ must also be nonnegative. This along with the fact that $\lim_{x\to\infty} \ell(x) = \infty$, we have that both $\lim_{x\to\infty} g(x) = \infty$ and $\lim_{x\to-\infty} g(x) = \infty$. Together, these imply that $\mathcal{L}^{unconstr}$ attains a minimum. Indeed, $\mathcal{L}^{unconstr}(0,0) = 3g(0)$, and there is some bound $B$ such that for all $x > B$ and $x < -B$, $g(x) > 3g(0)$. We can therefore restrict the optimization problem to $r_a, r_b \in [-B, B]$ without changing the solutions. Since $\mathcal{L}^{unconstr}$ is continuous and $[-B, B]^2$ is compact, a minimum is attained.

Next, note that $g$ is convex because compositions of convex functions with monotonic functions and convex combinations of convex functions are convex [24]. From this, we claim that if there is an optimal solution $(r_a, r_b)$, then $(r_a, r_a/2)$ is also an optimal solution. Indeed, fix such an $(r_a, r_b)$,

$$
\begin{aligned}
\mathcal{L}^{unconstr}(r_a, r_a/2) &= g(r_a - r_a/2) + g(r_a) + g(r_a/2) \\
&= g(r_a) + 2g(r_a/2) \\
&= g(r_a) + 2g(1/2(r_a - r_b) + 1/2 r_b) \\
&\leq g(r_a) + 2(1/2 g(r_a - r_b) + 1/2 g(r_b)) \\
&= \mathcal{L}^{unconstr}(r_a, r_b),
\end{aligned}
$$

where the inequality comes from convexity. This implies that $(r_a, r_a/2)$ is also optimal.

By above, we have that if $(r_b, r_a)$ is optimal, it must be the case that $r_a$ minimizes

$$
h(r_a) := 2g(r_a/2) + g(r_a).
$$

Observe that $h$ is again convex by monotonic composition and convex combinations.

Next, we will make use of the following facts about convex functions. Although they need not be differentiable, right- and left-hand derivatives always exist. For a function $k$ these are defined as

$$
k'_+(x) = \lim_{h\to 0^+} \frac{k(x+h) - k(x)}{h}
$$

$$
k'_-(x) = \lim_{h\to 0^-} \frac{k(x+h) - k(x)}{h}.
$$

Further, if $k$ is convex, we have that [24]:

(i)  $k'_+$ and $k'_-$ are nondecreasing,

(ii)  $k'_-(x) \leq k'_+(x)$  for every $x$,

(iii)  $x$ minimizes $k$ if and only if $k'_-(x) \leq 0 \leq k'_+(x)$,

(iv)  $k'_+$ and $k'_-$ are right and left continuous, respectively.

They also follow standard linearity and chain rule properties, which allow for simpler computation. For example:

if $k(x) = a\alpha(x) + b\beta(x)$, then $k'_+(x) = a\alpha'_+(x) + b\beta'_+(x)$,

if $k(x) = \alpha(\gamma x)$, then $k'_+(x) = \gamma\alpha'_+(\gamma x)$ if $\gamma \geq 0$, and $k'_+(x) = \gamma\alpha'_-(\gamma x)$ if $\gamma < 0$,

(all of these hold for left-hand derivatives by swapping the positions of $+$ and $-$) [9].

Next, we claim that we can find a valid $p$ (rational with $1/2 < p < 1$) and $w > 0$ such that $g'_+(w) > 0$, while $h'_+(w) < 0$. To that end, expanding the first derivative, we have

$$
g'_+(x) = -p\ell'_-(-x) + (1-p)\ell'_+(x).
$$

As long as $\ell'_-(-x) + \ell'_+(x) > 0$, this is strictly more than $0$ for $p$ satisfying

$$
p < \frac{\ell'_+(x)}{\ell'_-(-x) + \ell'_+(x)}. \tag{2}
$$

For $h$,

$$h'_+(x) = 2 \cdot 1/2 g'_+(x/2) + g'_+(x)$$
$$= -p\ell'_-(-x/2) + (1-p)\ell'_+(x/2) - p\ell'_-(-x) + (1-p)\ell'_+(x).$$
$$= -p[\ell'_-(-x/2) + \ell'_-(-x)] + (1-p)[\ell'_+(x/2) + \ell'_+(x)].$$

As long as $\ell'_-(-x/2) + \ell'_-(-x) + \ell'_+(x/2) + \ell'_+(x) > 0$, then this is strictly less than $0$ for

$$p > \frac{\ell'_+(x/2) + \ell'_+(x)}{\ell'_-(-x/2) + \ell'_-(-x) + \ell'_+(x/2) + \ell'_+(x)}. \tag{3}$$

Now, choose $w > 0$ such that $\ell'_-(-w/2) > \ell'_-(-w) \geq 0$. This is possible by using the following procedure. We know $\ell'_-(0) > 0$ (as otherwise $\ell(0) < \ell(x)$ for all $x < 0$, contradicting our assumption on $\ell$). We will split into cases depending on whether $\ell$ is nondecreasing or strictly-convex (at least one must be true by the theorem assumptions).

First, suppose $\ell$ is non-decreasing. This implies that $\ell'_-(x) \geq 0$ for all $x$. We know that there is a point $x < 0$ such that $\ell'_-(x) < \ell'_-(0)$ (as otherwise $\ell$ would eventually become negative). Let $d = \ell'_-(x)$. Take $w$ such that

$$-w = \sup\{x \mid \ell'_-(x) = d\}.$$

Note that $\ell'_-(-w) = d$ because $\ell'_-$ is left continuous (from (iv) above). Since $-w < 0$, so $-\frac{w}{2} > -w$. This implies that $\ell'_-(-\frac{w}{2}) > d$, as otherwise $-w$ would not be the supremum of such points. In addition, we have that $d \geq 0$ because $\ell$ is nondecreasing.

Next, suppose $\ell$ is strictly convex. then $\ell'_-$ is strictly increasing. Further, since it is left continuous and $\ell'_0(0) > 0$, there is a $\gamma > 0$ such that for all $x \in [-\gamma, 0]$, $\ell'_-(x) \geq 0$. Therefore, choosing $w = \gamma$ will do: $\ell'_-(-\gamma) \geq 0$ by choice of $\gamma$, and $-\gamma/2 > -\gamma$, so $\ell'_-(-\gamma/2) > \ell'_-(-\gamma)$ since $\ell'_-$ is strictly increasing.

For this choice of $w$, we first claim that the preconditions of denominators being positive hold for (2) and (3). Indeed, let us write $z_1, z_2, z_3, z_4$ for $\ell'_-(-2w), \ell'_-(-w), \ell'_+(w), \ell'_+(2w)$. We know that

$$z_1 \leq z_2 \leq z_3 \leq z_4$$

by properties of convexity, and since $\ell'_-(-w/2) > \ell'_-(-w) \geq 0$, we have that $0 \leq z_1 < z_2$. The denominators are of the form $z_1 + z_4$ and $z_1 + z_2 + z_3 + z_4$, which are now both necessarily positive. In addition, we also claim that a rational $p$ with $1/2 < p < 1$ satisfying both inequalities (2) and (3) will exist. Note that inequality (2) can now be represented as $\frac{z_4}{z_1 + z_4}$, and we have that

$$\frac{z_4}{z_1 + z_4} \leq 1$$

because $0 \leq z_1 < z_4$. Additionally, inequality (3) can be represented as

$$\frac{z_3 + z_4}{z_1 + z_2 + z_3 + z_4}$$

which is at least $1/2$ because $z_3 + z_4 > z_1 + z_2$. Finally, we have

$$\frac{z_3}{z_2 + z_3} \leq \frac{z_4}{z_2 + z_4} < \frac{z_4}{z_1 + z_4}$$

which implies that

$$\frac{z_3 + z_4}{z_1 + z_2 + z_3 + z_4} < \frac{z_4}{z_1 + z_4}.$$

Hence, there exists some rational $p$ in this interval, which is necessarily between $1/2$ and $1$, as needed.

To summarize, we have found a valid $p$ and value $w > 0$ such that $h'_+(w) < 0$ while $g'_+(w) > 0$. Because $h'_+$ is right continuous (from (iv) above), there is some $\gamma > 0$ such that $h'_+(w + \gamma) < 0$ as well. We will now show for all $(r_a, r_b) \in OPT^{unconstr}$, $r_a > w + \gamma$ and $r_b \leq w$. Indeed, note that $r_a$ must minimize $h$, so $h'_+(r_a) \geq 0$ (from (iii) above), which implies $r_a > w + c$. For $r_b$, suppose

for a contradiction $r_b > w$ as well. Note that $g'_+(w) > 0$ means $g$ is increasing to the right of $w$. Let $d = \min(r_a, r_b) - w > 0$. Consider $(r'_a, r'_b) = (r_a - d, r_b - d)$. We then have

$$
\begin{aligned}
\mathcal{L}^{unconstr}(r'_a, r'_b) &= g(r'_a - r'_b) + g(r'_a) + g(r'_b) \\
&= g(r_a - r_b) + g(r'_a) + g(r'_b) \\
&< g(r_a - r_b) + g(r_a) + g(r_b) \\
&= \mathcal{L}^{unconstr}(r_a, r_b).
\end{aligned}
$$

where the equality holds because $r'_a - r'_b = r_a - r_b$ and the inequality because $g$ is increasing to the right of $w$. Therefore, we reach a contradiction, because then $(r_a, r_b)$ would not be optimal. Thus, we have found values satisfying the lemma statement with $A_1 = w$ and $A_2 = w + \gamma$. $\qquad\square$

## A.2 Proof of Lemma 3.3

*Proof.* Fix $p$ inducing a nonempty $OPT^{unconstr}$ satisfying Lemma 3.2 with values $A_1$ and $A_2$. We first claim that for any $(r_a, r_b)$ (regardless of optimality), it is possible to find a $\theta$ such that $r_\theta(a) = r_a$ and $r_\theta(b) = r_b$. Indeed, note that

$$
\begin{pmatrix} 2 & 1 \\ 1 & 1 \end{pmatrix} \begin{pmatrix} \theta_1 \\ \theta_2 \end{pmatrix} = \begin{pmatrix} r_\theta(a) \\ r_\theta(b) \end{pmatrix}.
$$

Since $M = \begin{pmatrix} 2 & 1 \\ 1 & 1 \end{pmatrix}$ is invertible with inverse

$$
M^{-1} = \begin{pmatrix} 1 & -1 \\ -1 & 2 \end{pmatrix},
$$

for any $r_a, r_b$, we can simply set

$$
\theta = M^{-1} \begin{pmatrix} r_a \\ r_b \end{pmatrix}.
$$

Thus, $OPT^{com}$ is nonempty, and is simply the image of $OPT^{unconstr}$ under $M^{-1}$. Now, fix $(r_a, r_b) \in OPT^{unconstr}$, and let $\theta = M^{-1} \begin{pmatrix} r_a \\ r_b \end{pmatrix}$. By assumption, we have $r_a > A_2$ and $r_b \leq A_1$. This implies that $\theta_1 = r_a - r_b > A_2 - A_1$, while $\theta_2 = 2r_b - r_a < 2A_1 - A_2$. Thus, setting $A_3 = A_2 - A_1$ and $A_4 = 2A_1 - A_2$ satisfy the desired properties. $\qquad\square$

## A.3 Proof of Lemma 3.4

*Proof.* We first claim that $OPT(0) = OPT^{core}$. This will follow from showing

$$
\mathcal{L}^0(\theta) = 4 \cdot \mathcal{L}^{core}(\theta) + 3 \cdot \ell(0).
$$

Indeed, in $\mathcal{L}^0$, the copied candidates are in exactly the same location as their counterparts. Hence, each term in $\mathcal{L}^{core}$ appears 4 times, one for each combination of original and copy. In addition to these, there are the 6 terms for each ordered pair of $(a, a')$, $(b, b')$, and $(c, c')$. Note that, each $r_\theta(x) = r_\theta(x')$ for each $x \in \{a, b, c\}$ regardless of $\theta$ since they are in the same location. Therefore, the $\ell$ portion is always $\ell(0)$, and the corresponding $w_{x \succ x'}$ and $w_{x' \succ x}$ terms add up to one for each pair. Hence, the total sum of these terms is $3 \cdot \ell(0)$. Since $\mathcal{L}^0$ is equivalent to $\mathcal{L}^{core}$ up to positive scaling and translation, they have the same optima.

Finally, fix a $\theta \in OPT(0)$. Since $\theta \in OPT^{core}$, by Lemma 3.3, $\theta_1 > A_3$ and $\theta_2 < A_4$. Thus,

$$
r_\theta(c') = \varepsilon \cdot (-\theta_1 + \delta\theta_2) < \varepsilon(-A_3 + \delta A_4) < 0 = r_\theta(c)
$$

by choice of $\delta$. Hence, $\theta \in \overline{R^{c' \succ c}}$ $\qquad\square$

## A.4 Proof of Lemma 3.5

*Proof.* We first show that when optimizing each $\mathcal{L}^\varepsilon$, it is sufficient to consider only $\theta$ coming from a bounded region. Indeed, observe that $\mathcal{L}^\varepsilon(\mathbf{0}) = \binom{6}{2}\ell(0)$ for all $\varepsilon$. Since $\lim_{x \to \infty} \ell(x) = \infty$, we

can find some $B > 0$ such that for all $x > B$, $\ell(x) > \frac{\binom{6}{2}\ell(0)}{1-p} = \frac{\mathcal{L}^\varepsilon(\mathbf{0})}{1-p}$. For a pair of candidates $x \neq y \in C^{com}$, in the two terms concerning these candidates, we have

$$
\begin{aligned}
&w_{x \succ y}\ell(r_\theta(y) - r_\theta(x)) + w_{y \succ x}\ell(r_\theta(x) - r_\theta(y)) \\
&\geq (1-p)\left(\ell(r_\theta(y) - r_\theta(x)) + \ell(r_\theta(x) - r_\theta(y))\right) \\
&\geq (1-p)\ell(|r_\theta(y) - r_\theta(x)|).
\end{aligned}
$$

Applying this to $\{a, b\}$ and $\{b, c\}$,

$$
\begin{aligned}
\mathcal{L}^\varepsilon(\theta) &\geq (1-p)\left(\ell(|r_\theta(a) - r_\theta(b)| + \ell(|r_\theta(b)) - r_\theta(c)|)\right) \\
&= (1-p)(\ell(|\theta_1|) + \ell(|\theta_1 + \theta_2|))
\end{aligned}
$$

This implies that we may restrict our attention to $\theta$ in the region

$$
R^{bounded} = \{\theta \mid |\theta_1| \leq B, |\theta_2| \leq 2B\}.
$$

Indeed, for $\theta \notin R^{bounded}$, either $|\theta_1| > B$ or $|\theta_1 + \theta_2| > B$. In either case, we have $\mathcal{L}^\varepsilon(\theta) \geq (1-p)\ell(B) > \mathcal{L}^\varepsilon(\mathbf{0})$.

Note that $\mathcal{L}^\varepsilon(\theta)$ is continuous not only in $\theta$, but also in $\varepsilon$. Additionally, $R^{bounded}$ is closed and bounded, and hence, compact. Therefore, by Berge's Maximum Theorem, $OPT(\varepsilon)$ is nonempty and *upper semi-continuous* in $\varepsilon$ [3]. As per the definition of upper semi-continuous, since $OPT(0) \subseteq \overline{R^{c' \succ c}}$, an open set, for sufficiently small $\varepsilon > 0$, $OPT(\varepsilon) \subseteq \overline{R^{c' \succ c}}$. Finally, note that $R^{c' \succ c} \cap R^{bounded}$ is compact, so a minimum is attained, and this minimum must therefore be strictly larger than the values attained by members of $OPT(\varepsilon)$. □

## A.5 Proof of Theorem 3.6

*Proof.* Without loss of generality, we may assume that $\inf_x \ell(x) = 0$, as otherwise we could translate $\ell$ without affecting the optimization problem. Fix a profile $\pi$ with feasible PMC ranking $\sigma$, and let $\theta^{PMC}$ be a non-degenerate parameter vector that induces $\sigma$.

First, we show that $\inf_\theta \mathcal{L}^{maj}(\theta; \pi, \ell) = 0$. Indeed, note that $c \cdot \theta^{PMC} \in R^\pi$ for all $c > 0$. Further, note that for any $a, b$ with $w_{a \succ b}(\pi) > 1/2$, $r_{\theta^{PMC}}(b) - r_{\theta^{PMC}}(a) < 0$. Therefore, by making $c$ large, the nonzero terms in $\mathcal{L}^{maj}$ will have an input to $\ell$ negative and becoming arbitrarily large in magnitude. Since $\ell$ is nondecreasing, these approach the infimum of 0.

Next, for any $\sigma' \neq \sigma$, $\inf_\theta \mathcal{L}^{maj}(\theta; \pi, \ell) \geq \ell(0)$. Indeed, there must be some pair of candidates $a, b$ with $a \succ_\sigma b$ and $b \succ_{\sigma'} a$. For any $\theta \in R^{\sigma'}$, $r_\theta(b) \geq r_\theta(a)$, so $\ell(r_\theta(b) - r_\theta(a)) \geq \ell(0)$, and this lower bounds the loss function. □

## A.6 Proof of Theorem 3.7

*Proof.* We construct an explicit instance and pairwise majority relationships such that no matter what feasible ranking a rule picks, there is an underlying profile where that output was a PO violation.

We will have 9 candidates; 8 will be labeled $c_i^+$ and $c_i^-$ for $i = 1, 2, 3, 4$, and one labeled $c^*$. They will have feature vectors in $\mathbb{R}^4$. Each $c_i^\pm$ will be located at $\mathbf{x}_{c_i^\pm} = \pm e_i$ where $e_i$ is the $i$'th standard basis vector, i.e., $c_2^+$ is at $(0, 1, 0, 0)$ and $c_4^-$ is at $(0, 0, 0, -1)$. Finally, $c^*$ will be located at $(1/5, 1/5, 1/5, 1/5)$.

There will be 5 voters. Their pairwise majority graph will be as follows. Candidate $c^*$ will pairwise beat all others. In addition, each $c_i^+$ will pairwise beat each $c_j^-$. Among the $c_i^+$ candidates, there will be a cycle $c_1^+ \succ c_2^+ \succ c_3^+ \succ c_4^+ \succ c_1^+$ and between the remaining two pairs $c_1^+ \succ c_3^+$ and $c_2^+ \succ c_4^+$. The $c_i^-$ candidates will be the exact reverse of this, i.e., a cycle $c_4^- \succ c_3^- \succ c_2^- \succ c_1^- \succ c_4^-$, along with $c_3^- \succ c_1^-$ and $c_4^- \succ c_2^-$. A pictorial representation can be found in Figure 1.

A C1 rule must pick a $\theta$ solely based on the pairwise majority graph. We will show that regardless of what $\theta$ it outputs, this will lead to a PO violation.

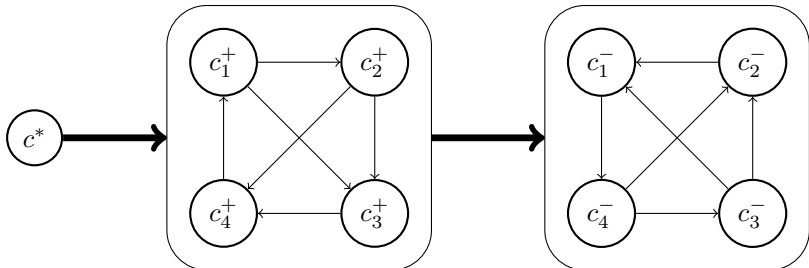

Figure 1: Graph showing pairwise majority relationship between candidates. Regular edges show relationships among $c_i^+$ candidates and among $c_i^-$ candidates. Thick edges indicate that $c^*$ pairwise beats all candidates, and each $c_i^+$ pairwise beats each $c_j^-$ candidate.

To that end, the first fact we will show is that no $\theta$ can rank $c^*$ first. Indeed, for any $\theta$, $r_\theta(c^*) = \frac{1}{5}\sum_i \theta_i \leq \frac{1}{5}\sum_i |\theta_i| < \max_i |\theta_i|$. On the other hand, for $i$ maximizing $|\theta_i|$, at least one of $c_i^{\pm}$ will achieve this reward, strictly larger than $r_\theta(c^*)$. Hence, regardless of the output $\theta$, some candidate must be ranked above $c^*$. We will show that this leads to a PO violation.

To construct profiles consistent with the pairwise majority graph, voters will always have rankings of the following form:

$$c_i^+ \succ c^* \succ c_j^+ \succ c_k^+ \succ c_\ell^+ \succ c_\ell^- \succ c_k^- \succ c_j^- \succ c_i- \tag{4}$$

for some $\{i, j, k, \ell\} = \{1, 2, 3, 4\}$. In other words, they will rank a single $+$ candidate above $c^*$ and the rest in some order, followed by all $-$ candidates in the reverse order. This is always achievable with the voter vector that puts values $1, 3\varepsilon, 2\varepsilon, 1\varepsilon$ in entires $i, j, k,$ and $\ell$, respectively, for some small $\varepsilon > 0$.

Fix an output $\theta$ with induced ranking $\sigma$. There must be at least one candidate $c \neq c^*$ ranked above $c^*$. We now split into cases depending on which candidate this is. For each choice, we will construct a profile consistent with the pairwise majority graph where the candidate above $c^*$ is Pareto dominated by $c^*$. We will describe each voter's ranking only by an ordering over the $+$ candidates, assuming they otherwise take the form described in (4). Note that $c^*$ is always ranked second, so if a candidate is never ranked first, they are Pareto dominated by $c^*$. The profiles for each candidate $c_i^+$ can be found in the following table. One can check that all pairwise relationships are satisfied, and the corresponding $c_i^+$ is never ranked first.

Table 1: Profiles with 5 voters and consistent with the pairwise majority graph where where the corresponding candidate is PO-dominated by $c^*$. The notation $1 : (2, 1, 3, 4)$ implies one voter has the ranking in the form of (4) with $(i, j, k, \ell) = (2, 1, 3, 4)$.

| $c_1^+$ | $c_2^+$ | $c_3^+$ | $c_4^+$ |
|---|---|---|---|
| 1: $(2, 1, 3, 4)$ | | | |
| 1: $(3, 1, 2, 4)$ | 2: $(1, 2, 3, 4)$ | 2: $(1, 2, 3, 4)$ | 2: $(1, 2, 3, 4)$ |
| 1: $(3, 2, 4, 1)$ | 2: $(3, 2, 4, 1)$ | 2: $(2, 3, 4, 1)$ | 2: $(2, 4, 1, 3)$ |
| 2: $(4, 1, 2, 3)$ | 1: $(4, 1, 2, 3)$ | 1: $(4, 1, 2, 3)$ | 1: $(3, 4, 1, 2)$ |

Finally, if a $-$ candidate is ranked above $c^*$, then any of the following profiles work, as all $-$ candidates are Pareto dominated by $c^*$ with rankings shown in (4). □

## B  PMC Infeasibility Example

Consider the case with $d = 3$ and seven candidates: one special candidate $a^*$ located at $(1/4, 1/4, 1/4)$, and six others $c_i^{\pm}$ located at standard basis vectors $e_i^{\pm}$. We have three voters with parameter vectors $(1, 2\varepsilon, \varepsilon)$, $(2\varepsilon, 1, \varepsilon)$, and $(2\varepsilon, \varepsilon, 1)$, where $\varepsilon < 1/5$ is a small positive number. These voters have the following induced rankings:

| Rank | $v_1$ | $v_2$ | $v_3$ |
|------|-------|-------|-------|
| 1 | $c_1^+$ | $c_2^+$ | $c_3^+$ |
| 2 | $a^*$ | $a^*$ | $a^*$ |
| 3 | $c_2^+$ | $c_1^+$ | $c_1^+$ |
| 4 | $c_3^+$ | $c_3^+$ | $c_2^+$ |
| 5 | $c_3^-$ | $c_3^-$ | $c_2^-$ |
| 6 | $c_2^-$ | $c_1^-$ | $c_1^-$ |
| 7 | $c_1^-$ | $c_2^-$ | $c_3^-$ |

We argue the ranking $a^* \succ c_1^+ \succ c_2^+ \succ c_3^+ \succ c_3^- \succ c_2^- \succ c_1^-$ is a PMC ranking but no linear reward function can position $a^*$ at the top of this ranking.

For any reward vector $\theta = (\theta_1, \theta_2, \theta_3) \in \mathbb{R}^3$, the reward for $a^*$ is :

$$r_{a^*}^\theta = \frac{1}{4}(\theta_1 + \theta_2 + \theta_3)$$

Given the placement of $c_i^\pm$ at the standard basis vectors, each $c_i^\pm$ achieves a reward equivalent to one of the absolute values of the components of $\theta$, thus surpassing $r_{a^*}^\theta$ since

$$r_{a^*}^\theta < \max_i |\theta_i|.$$

## C  Additional Axiomatic Properties of Social Choice-Based Rules

We begin by stating another prominent axiom from social choice theory.

**Definition C.1** (Separability). *A ranking aggregation rule satisfies* ranking separability *(or* separability *for short) if, when two profiles yield identical output rankings, when combined into a single profile, this should also produce the same output ranking.*

Ranking separability preserves consistency in aggregation outputs and ensures stable decisions across similar preference distributions.

### C.1  Copeland Violates Separability

**Theorem C.2.** *Both Copeland (in traditional social choice) and LCPO (in linear social choice) fail separability.*

*Proof.* Consider the following two profiles on 7 candidates with five and three voters each:

| Rank | $v_1$ | $v_2$ | $v_3$ | $v_4$ | $v_5$ |
|------|-------|-------|-------|-------|-------|
| 1 | $a$ | $b$ | $b$ | $c$ | $d$ |
| 2 | $g$ | $a$ | $a$ | $e$ | $c$ |
| 3 | $d$ | $c$ | $e$ | $f$ | $f$ |
| 4 | $e$ | $e$ | $d$ | $g$ | $g$ |
| 5 | $f$ | $d$ | $f$ | $b$ | $b$ |
| 6 | $b$ | $g$ | $g$ | $a$ | $e$ |
| 7 | $c$ | $f$ | $c$ | $d$ | $a$ |

| Rank | $v_6$ | $v_7$ | $v_8$ |
|------|-------|-------|-------|
| 1 | $a$ | $b$ | $c$ |
| 2 | $d$ | $a$ | $a$ |
| 3 | $b$ | $e$ | $e$ |
| 4 | $f$ | $d$ | $b$ |
| 5 | $c$ | $c$ | $f$ |
| 6 | $g$ | $g$ | $g$ |
| 7 | $e$ | $f$ | $d$ |

In the first profile, the Copeland scores are $5, 4, 3, 3, 3, 2, 1$ for candidates $a, b, c, d, e, f, g$, respectively. Similarly, in the second profile, they are $6, 5, 3, 3, 3, 1, 0$. So under any consistent tie-breaking rule, both of these profiles would output the ranking $a \succ b \succ c \succ d \succ e \succ f \succ g$.

However, if we combine these two profiles, then the score of $a$ is $5$ while the score of $b$ is $6$, and thus, $b$ will be ranked above $a$, violating separability.

To see that this also holds for LCPO, note that when every ranking is feasible, LCPO coincides with Copeland. We can simply have 7 candidates in $\mathbb{R}^7$ all located at unit vectors, and voters with inputs from the above profiles. $\square$

## C.2 Linear Kemeny Rule

Next, we consider a different rule from social choice theory, the Kemeny rule, which can be transformed to the linear setting while maintaining the separability can be achieved along with PMC. Given an input profile $\pi$, the Kemeny rule returns a ranking $\sigma^*$ that minimizes the total number of pairwise disagreements with voters rankings, i.e.

$$\sigma^* \in \arg\min_{\sigma \in S^m} \sum_{i \in V} \sum_{(a,b): a \succ_{\sigma_i} b} \mathbb{1}(b \succ_\sigma a)$$

where $S^m$ contains all possible permutations of the $m$ candidates. This expression can be equivalently written as

$$\sigma^* \in \arg\min_{\sigma \in S^m} \sum_{a \neq b \in C} n_{a \succ b}(\pi) \cdot \mathbb{1}(b \succ_\sigma a).$$

Here, we define the *linear Kemeny rule*, which outputs a parameter vector $\theta^*$ that induces a ranking that minimizes the total number of pairwise disagreements with voters' rankings, i.e.,

$$\theta^* \in \arg\min_{\theta \in \mathbb{R}^d} \sum_{a \neq b \in C} n_{a \succ b}(\pi) \cdot \mathbb{1}(r_\theta(b) > r_\theta(a)).$$

Note that this rule conforms to the standard loss formulation, where the loss function is binary: it is $0$ if two rankings agree with respect to the relative ranking of a pair of candidates and $1$ otherwise. Since binary loss is not convex, it does not fit in the impossibility result of Theorem 3.1.

Note that Kemeny is generally NP-hard to compute; however, even for linear Kemeny, there is at least an exponential time aglorithm by brute-force computing the score of every ranking, and determining whether or not it is feasible.

**Theorem C.3.** *Linear Kemeny satisfies PMC and separability.*

*Proof.* PMC holds because the linear Kemeny score minimizes disagreements even among non-transitive rankings, making the PMC ranking the optimal choice whenever it is feasible. Separability is evident as the Kemeny score of a ranking over two datasets is simply the sum of the scores in each dataset. If the same ranking minimizes the score in both datasets independently, it will also minimize the score in their combination. □

**Theorem C.4.** *Linear Kemeny does not satisfy PO or majority consistency.*

*Proof.* Consider the scenario with 20 candidates whose feature vectors are represented in the table below:

| Candidate | Feature Vector |
|---|---|
| 1 | $(2000000, 0, 0, 0, 0, 0, 0)$ |
| 2 | $(0, 2000000, 0, 0, 0, 0, 0)$ |
| 3 | $(0, 200000, 0, 0, 0, 0, 0)$ |
| 4 | $(0, 100000, 100000, 0, 0, 0, 0)$ |
| 5 | $(0, 0, 200000, 0, 0, 0, 0)$ |
| 6 | $(0, 0, 20000, 0, 0, 0, 0)$ |
| 7 | $(0, 0, 10000, 10000, 0, 0, 0)$ |
| 8 | $(0, 0, 0, 20000, 0, 0, 0)$ |
| 9 | $(0, 0, 0, 2000, 0, 0, 0)$ |
| 10 | $(0, 0, 0, 1000, 1000, 0, 0)$ |
| 11 | $(0, 0, 0, 0, 2000, 0, 0)$ |
| 12 | $(0, 0, 0, 0, 200, 0, 0)$ |
| 13 | $(0, 0, 0, 0, 100, 100, 0)$ |
| 14 | $(0, 0, 0, 0, 0, 200, 0)$ |
| 15 | $(0, 0, 0, 0, 0, 20, 0)$ |
| 16 | $(0, 0, 0, 0, 0, 10, 10)$ |
| 17 | $(0, 0, 0, 0, 0, 0, 20)$ |
| 18 | $(0, 0, 0, 0, 0, 0, 2)$ |
| 19 | $(1, 0, 0, 0, 0, 0, 1)$ |
| 20 | $(2, 0, 0, 0, 0, 0, 0)$ |

Each vector is constructed such that candidates are prioritized based on the magnitude of their first non-zero entry, leading to a natural ordering within grouped subsets: $\{1, 2\} \succ \{3, 4, 5\} \succ \{6, 7, 8\} \succ \{9, 10, 11\} \succ \{12, 13, 14\} \succ \{15, 16, 17\} \succ \{18, 19, 20\}$. We will have six voters, with rankings induced by the parameter vectors described below.

| Voter | Parameter Vector |
|-------|------------------|
| $v_1$ | $(2, 1, 7, 6, 5, 4, 3)$ |
| $v_2$ | $(3, 2, 1, 7, 6, 5, 4)$ |
| $v_3$ | $(4, 3, 2, 1, 7, 6, 5)$ |
| $v_4$ | $(5, 4, 3, 2, 1, 7, 6)$ |
| $v_5$ | $(6, 5, 4, 3, 2, 1, 7)$ |
| $v_6$ | $(7, 6, 5, 4, 3, 2, 1)$ |

Each voter ranks candidates within the group in increasing order except for a single reversed group. For instance, $v_1$ ranks candidates as $1 \succ 2 \succ \mathbf{5} \succ \mathbf{4} \succ \mathbf{3}$ and so forth. Under this setup, no parameter vector $\theta$ can create a ranking that satisfies all voters' preferences due to the cyclic nature and individual group preferences.

We can check that linear Kemeny rule outputs a ranking in which candidate 2 is ranked above 1. From this analysis, we also conclude that the rules does not satisfy majority consistency either. $\square$

A possibly easy fix to the problem that linear Kemeny does not satisfy PO would be to enforce the PO criterion, as we did for the LCPO, i.e., to restrict to parameter vectors that respect PO. However, we show that if we do that, then linear Kemeny subject to PO does not satisfy separability anymore.

**Theorem C.5.** *Linear Kemeny subject to PO violates separability and majority consistency.*

*Proof.* First, consider a set of candidates and a profile similar to the one that is given in the proof of Theorem C.4. When we restrict to reward functions that 1 above 2, then we can check that Linear Kemeny subject to PO outputs one of the rankings that are in the input profile. Without loss of generality, assume that it outputs the ranking of $v_1$.

Second, consider the same set of candidates and three voters, with rankings induced by the following parameter vectors:

| Voter | Parameter Vector |
|-------|------------------|
| $v_1'$ | $(2, 1, 7, 6, 5, 4, 3)$ |
| $v_2'$ | $(2, 1, 7, 6, 5, 4, 3)$ |
| $v_3'$ | $(1, 7, 6, 5, 4, 3, 2)$ |

In this case, Linear Kemeny subject to PO outputs the ranking of $v_1'$ which is the same with this of voter $v_1$.

When the two profiles are combined, then we do not anymore restrict on rankings in which 1 is above 2, since 1 does not Pareto dominates 2 anymore. Then, we can check that linear Kemeny subject to PO outputs a different ranking than before in which 2 is ranked above 1. From this example, we see that linear Kemenery subject to PO still violates majority consistency. $\square$

### C.3 Leximax Plurality

Plurality is probably the most ubiquitous voting rule in the world. Its ranking variant ranks the candidates in decreasing order with respect to their *plurality scores*. The plurality score of a candidate is equal to the number of her appearances in the first position. This rule is known to satisfy several axioms but in linear social choice cannot be directly applied, as not all rankings are feasibly.

Similarly to leximax Copeland, we define Leximax Plurality as follows. It ranks first the candidate with the highest plurality score that can be ranked first under some parameter vector. Subject to this first position, it ranks second the candidate with the highest plurality score that can feasibly be ranked second, and so on, until all the positions are filled.

**Theorem C.6.** *Leximax Plurality satisfies majority consistency, winner monotonicity and separability.*

*Proof.* Note that leximax Plurality always returns a ranking in which the candidate with the highest plurality score is ranked first, since there exists at least one feasible ranking in which this candidate is ranked first. From this observation, we immediately see that leximax Plurality satisfies majority consistency.

Now, suppose that on input profile $\pi$, the rule outputs a ranking such that candidate $a$ is ranked first. This means that $a$ has the highest plurality score. Now, consider a profile $\pi'$ which is similar to $\pi$ with the only exception being a ranking in which $a$ is ranked in a higher position. It is clear that $a$ continues to have the highest plurality score and therefore leximax Plurality will output a ranking in which $a$ is ranked first. Therefore, winner monotonicity is satisfied. .

It remains to prove that the rule satisfies separability. Suppose that under two different profiles $\pi_1$ and $\pi_2$, the rule outputs $\sigma$, and under the aggregated profile $\pi_3$, it outputs $\sigma'$. We will show that $\sigma = \sigma'$. One main observation is that if a candidate $a$ has a higher plurality score than a candidate $b$ under both $\pi_1$ and $\pi_2$, then $a$ has a higher plurality score than $b$ under $\pi_3$ as well. We will show the desired property by induction on the positions of the ranking $\sigma$. Start from the first position in which say candidate $a$ is ranked first. From above, we know that $a$ has the highest plurality score under both $\pi_1$ and $\pi_2$, which remains true in $\pi_3$, and therefore $a$ is ranked first in $\sigma'$. Now, assume that up to position $t-1$, $\sigma$ and $\sigma'$ are similar and denote with $a'$ the candidate that is ranked at position $t$ of $\sigma$. We denote by $S_1$, $S_2$ and $S_3$ the set of candidates that are not ranked among the first $t$ positions in $\sigma$ and have higher plurality score than $a'$ under $\pi_1$, $\pi_2$, and $\pi_3$ respectively. Since, $a'$ is ranked at the $t$-th postion in $\sigma$, we get that, subject to the fixed first $t-1$ position, no candidate in $S_1 \cup S_2$ can be ranked at the $t$-th position. The theorem follows by noticing that $S_3 \subseteq S_1 \cup S_2$, which follows from the main observation above. $\qquad \square$

