# OpenReview forum: "Axioms for AI Alignment from Human Feedback"
_NeurIPS.cc/2024/Conference — NeurIPS 2024 spotlight_

### Official Review · Reviewer_Gde6 · 2024-06-15

**Soundness:** 3
**Presentation:** 3
**Contribution:** 3
**Rating:** 7
**Confidence:** 1

**Summary:**

Authors argue that for RLHF, the preferences are pairwise and we need to train a model that respects the preferences in aggregate which is in scope of social choice theory. Then they evaluate different aggregation methods on if they respect well established axioms. They showed that the popular TB model does not respect some axioms and came up with novel rules for reward learning.

**Strengths:**

- principled approach to model preferences from population, rooted in social choice theory
- relevant topic for this conference
- initated the filed to approach RLHF reward modeling to follow axiomatic guidance through social choice theory

**Weaknesses:**

- only theoretical contributions, we do not know how the theory translates to in practice, even in toy settings

**Questions:**

- could you cite the places where you obtained the axioms of social choice theory?
- since you did not include where these axioms are from, are you missing any investigations of other axioms? If so, why did you choose the ones you investigated and why leaved the others out?

**Limitations:**

written in the discussion section

---

> ### Author Rebuttal · Authors · 2024-08-06
>
> > Could you cite the places where you obtained the axioms of social choice theory?
>
> We consider canonical axioms from the social choice literature. See, for instance, "The Handbook of Computational Soical Choice," Chapter 2. We are happy to add further references in a revision.
>
> > Since you did not include where these axioms are from, are you missing any investigations of other axioms? If so, why did you choose the ones you investigated and why leaved the others out?
>
> Indeed, there are a great number of axioms in the social choice literature one could consider. Our choice of PMC and PO was due to these serving as very natural starting points for this kind of investigation. In particular, Pareto optimality is viewed as one of the most fundamental properties expected of social choice rules, and is satisfied by practically all reasonable social choice rules, whereas PMC related to Condorcet consistency, one of the most extensively studied axioms. We also considered several others (see some discussion of these in Appendix B). In any case, we believe that an extensive study of other social choice axioms is an important subject for follow-up work.

---

> > ### Comment · Reviewer_Gde6 · 2024-08-08
> >
> > The authors have adequately addressed all of my concerns. I will keep my original positive score.

---

### Official Review · Reviewer_6JKc · 2024-07-08

**Soundness:** 3
**Presentation:** 3
**Contribution:** 3
**Rating:** 6
**Confidence:** 3

**Summary:**

Recent months have seen a flood of concurrent papers studying the relationship between RLHF, preference aggregation, and social choice theory. This paper joins these lines of work and studies how to aggregate diverse human preferences (in the context of RLHF) that are modeled as a random utility model (e..g., BTL). The authors adopt a social choice perspective and show that the BTL model (and similar ones) fail to satisfy basic axioms known from social choice theory. The paper then also proposes a leximax Copeland (subject to PO) rule that satisfies desirable properties such as pareto optimality and majority consistency.

**Strengths:**

- The paper is very well-written.
- The studied problem is relevant and timely.
- The authors do a good job explaining concepts from social choice theory so that the paper is also easy to read for researchers who don't have a background in computational social choice.
- While unsurprising, the main theoretical result that linear aggregation (using a linearly parameterized reward function) is insufficient to achieve basic axioms of "fair" and reasonable preference aggregation is useful and interesting.

**Weaknesses:**

- The assumption that for each voter the complete ranking is available is obviously unrealistic, which limits the applicability of the proposed leximax Copeland rule.
- I believe that this paper's practical relevance is quite limited. It is unclear how leximax Copeland could be implemented in practice. The sampling strategy over voters and candidates (prompt/responses) is far from obvious and not discussed in the paper. Any additional comments about how such a sampling strategy would look like would be appreciated.
- In contrast to the various related works on RLHF + social choice, which provide algorithms/solutions to augment the traditional RLHF framework, this work does not provide a tractable approach to address the problem of aggregating preferences in a reasonable way.

**Questions:**

- What about non-linear reward function classes?

**Limitations:**

The authors address the limitations of the work adequately in Section 5.

---

> ### Author Rebuttal · Authors · 2024-08-06
>
> > The sampling strategy over voters and candidates (prompt/responses) is far from obvious and not discussed in the paper. Any additional comments about how such a sampling strategy would look like would be appreciated.
>
> We believe you are referring to this passage in the discussion: "However, the complete rankings are not necessary for computing this rule, rather, all we need to know are PO dominance relationships and majority directions. We can therefore apply the rule whenever we can determine this information at least approximately through, e.g., sampling."
>
> We had the following model in mind: Each participant is shown a certain number of randomly selected pairs of candidates. We should be able to estimate all pairwise margins within a small error as long as each pair is compared at least $\Omega(\log m)$ times in expectation.
>
> The only information needed to compute LCPO is for each pair of candidates in the dataset (i) which a majority of voters prefer and (ii) whether all voters have this preference. Determining these exactly is impossible with only approximate pairwise margins. However, running LCPO on the estimated margins would get an "approximate" LCPO winner, in the sense that the output ranking would be the winner on a profile differing on only a small fraction of voters.
>
> > What about non-linear reward function classes?
>
> This is a great question. We first note  that linear reward models are more general than they may first appear, as the "feature vectors" can be output embeddings of a pretrained LLM (and we just learn a linear layer of a reward model from pairwise comparisons). Consequently, even a linear model can in principle encode highly non-linear characteristics of the raw input (e.g., text).
>
> Nonetheless, expanding beyond linear reward functions is an important direction for future work. A significant challenge is that once we consdier nonconvex model classes (e.g., even small multi-layer perceptrons), they seems difficult to analyze theoretically, as we now run into an issue of local optima and computational intractability of computing globally optimal solutions.

---

> > ### Comment · Reviewer_6JKc · 2024-08-07
> >
> > Thanks for your response. I'm raising my score to a 6. I think, at least in terms of numerical score, my original assessment was slightly too harsh.
> >
> > I guess my main concern is still that there are so many concurrent and earlier papers addressing the same problem, which somewhat dillutes the contributions of this work. Nevertheless, this is solid, well-presented work, which I'm in favor of accepting.

---

### Official Review · Reviewer_xjPu · 2024-07-10

**Soundness:** 4
**Presentation:** 3
**Contribution:** 4
**Rating:** 8
**Confidence:** 3

**Summary:**

The paper presents a axiomatic social choice framework for the problem of doing RLHF on group preferences. It shows that classical RLHF (and indeed a wider class of similar methods) violates the Pareto Optimality and Pairwise Majority Consistency axioms, and shows (via explicit construction) that there are mechanisms that satisfy both axioms along with two other axioms. (To simplify the model, the space of preferences is assumed to be implementable via linear classification.)

**Strengths:**

**High importance and novelty**
- LLMs, and also AI systems in general, are expected to have large societal impact now and in the future. In light of this expectation, how to make sure model policies fairly represent the welfare of all stakeholders is important. This question, to my knowledge, has not received sufficient analysis except in this work and some of its concurrent works.

**Flawless construction of the theoretical framework**
- The theoretical setup seems flawless, covering exactly all the fundamental components of the problem, and in a very elegant manner.

**Weaknesses:**

I have some suggestions for improvement/future work, but I don't feel like they count as "weaknesses" *per se*, since those are very high standards which I don't feel like a normal conference paper would be held up to. I am instead putting those suggestions in the limitations section.

Also, please note that I did not check the proofs.

**Questions:**

- Do you think the PO and PMC axioms (possibly along with majority consistency, winner monotonicity etc.) are in any sense the "gold standards"? Could there be other similarly reasonable axioms that are contradictory with PO/PMC? I'm asking this because I have the intuition (which could be wrong) that an Arrow-like impossibility result would also haunt the linear social choice setting; elaboration in the limitations section.

**Limitations:**

I highly appreciate the detailed discussion of limitations and future directions in the Discussion section of the paper, and I agree with most of the points there. I have the following two additional remarks, which are meant not as critiques but as suggestions for exploration.

1. **(Non)existence of a gold standard in linear social choice**
    - Let's look at Theorem 3.1 first. If the RLHF method decide to overturn a perfect consensus on one pair with a tiny margin, in order to much more significantly reduce loss on a divisive pair (which is possible due to strict convexity), this actually seems desirable, despite violating PO.
        - One could counter that whether the margin is small or large does not matter when what we care about the outcome is only the ordering. However, the exact margins (<> the exact reward values) do matter a lot in practice, where they indirectly (in RLHF) or directly (in DPO) determines what probabilities to assign to each response (as opposed to merely a ranking relation between the responses).
    - In general, I suspect there is some arrow-like impossibility result in this space, where all the desirable properties just cannot be met at the same time. It's unclear which among the conflicting properties (somewhere among them are PO and PMC, and somewhere else is "prioritize preference violations with larger margins", with many others) are the most important.
    - This intuition seems to be confirmed by the LCPO construction, which introduces rather arbitrary requirements (namely hard-coding the PO rule into the LCPO algorithm) in order to satisfy PO.
2. **Human evaluations and human subject experiments**
    - I suspect that human evaluation (e.g. letting human subjects judge the fairness of preference aggregation outcomes) could be an equally, if not more, important criteria for preference aggregation mechanisms than formal axioms are, given that (1) it's unclear which formal principles are more important than others and (2) these principles could be in conflict with each other (as in Arrow's theorem).
    - This could be used to evaluate both aggregation mechanisms (does the outcomes align with human judgment?) and axioms (is human judgment in line with this axiom in general? If not, why, and who is right?).
    - Here, the line between computer science and cognitive science seems to be dissolving.

---

> ### Author Rebuttal · Authors · 2024-08-06
>
> We appreciate your discussion of limitations. You raise excellent points on the challenges here (and some relate to discussions we have had amongst ourselves!). With regard to your direct questions:
>
> > Do you think the PO and PMC axioms (possibly along with majority consistency, winner monotonicity etc.) are in any sense the "gold standards"?
>
> We view both PO and PMC as relatively minimal requirements, more so than majority consistency and winner monotonicity. For example, PO (at least in traditional social choice) is a basic property that is satisfied by essentially all natural rules. Nevertheless, we agree that many other axioms can be considered, and it is indeed not self-evident that enforcing PO outweighs the tradeoff of not respecting large margins. We do not view our axioms and results as the final word. Rather, we hope the paper offers a starting point for having such conversations.
>
> > Could there be other similarly reasonable axioms that are contradictory with PO/PMC?
>
> This seems quite plausible. We were quite surprised by the fact that PO and C1 are incompatible given how easy they are to achieve in traditional social choice. It does suggest that there may be other strong impossibilities yet to be discovered.

---

> ### Comment · Reviewer_xjPu · 2024-08-07
>
> I appreciate the authors' response. The authors' replies to my questions are reasonable, and I strongly encourage the authors to include these discussions in the paper. I will leave my original score unchanged.

---

### Official Review · Reviewer_TLca · 2024-07-13

**Soundness:** 3
**Presentation:** 4
**Contribution:** 2
**Rating:** 6
**Confidence:** 3

**Summary:**

The paper proposes an axiomatic approach to study preference aggregation for AI alignment. Inspired by works in social choice theory, the authors investigate a paradigm that they call linear social choice where preferences are representable by a linear model. In this context, they notably prove that if the linear rank aggregation rule minimizes some natural loss function (like the one induced by the Bradley-Terry model), then it cannot satisfy Pareto optimality. To circumvent this issue, the authors propose a variation of Copeland rule that outputs a ranking representable by a linear model.

**Strengths:**

The paper investigates an important research question with a theoretically-founded approach.

The obtained results seem to be novel as far as I know. The failure of PO by minimizing a loss points to a strong limitation of the current approaches using human feedback.

The paper is well and clearly written. I notably appreciate the footnotes, comments, and connections with other works that the authors make.

**Weaknesses:**

I feel that the linear social choice paradigm may be too restrictive. I believe that the authors focus on linear models, because this is indeed a natural machine learning model. However, I don't think it is necessary to assume that the rankings of voters (i.e., humans in AI alignment) have to be linearly representable. In my opinion, the paper would be stronger if the results were presented without this latter assumption.

There are some gaps between the setting in machine learning (e.g., RLHF) and social choice theory. As the authors mention, in the latter, one usually observes pairwise comparisons instead of rankings. In addition, I think in the latter, one may not be interested to recover the full ranking, but only part of it, e.g., in RLHF, as the agent is trained, the reward approximator only needs to be good in the region of state-action pairs visited by a good policy. Could the authors comment on how this would impact their results?

The proposed social choice-based rule seems to me to be a bit artificial and only created to enforce Pareto optimality. It is not clear if such rule could easily be implemented in RLHF for instance and obtained via learning. For instance, in RLHF, this rule would require to consider all trajectories, which is impractical for any complex problem.

**Questions:**

1) Could the authors comment on how only trying to recover part of the full ranking would impact their results?

2) Could you the authors comment on their proposed rule could be implemented in practice?

**Limitations:**

The current discussion is adequate.

---

> ### Author Rebuttal · Authors · 2024-08-06
>
> > Could the authors comment on how only trying to recover part of the full ranking would impact their results?
>
> We believe our work addresses this concern. The prompt/response datasets are derived from sampling an LLM, ensuring the corresponding feature vectors are already in a reasonable region of feature space simply by virtue of being in the dataset. Our axioms are designed to guarantee good performance specifically on these vectors.
>
> We could have instead chosen even stronger notions requiring the conditions to hold across the entire feature space. For instance, PO could be defined such that for *any* two feature vectors $\mathbf{x}, \mathbf{x}' \in \mathbb{R}^d$, if all voters prefer $\mathbf{x}$ to $\mathbf{x}'$, then $\theta^*$ should as well. This is (i) impossible to achieve with trivial counterexamples, and (ii) unnecessarily stringent for precisely the reasons you mentioned.
>
> This idea extends to RLHF beyond LLMs, as long as the dataset defines a reasonable region for the reward approximator to perform well in.
>
> > The proposed social choice-based rule seems to me to be a bit artificial and only created to enforce Pareto optimality. It is not clear if such rule could easily be implemented in RLHF for instance and obtained via learning. For instance, in RLHF, this rule would require to consider all trajectories, which is impractical for any complex problem. [...] Could you the authors comment on their proposed rule could be implemented in practice?
>
> We believe that the problem you're alluding to may not be an issue. In our desired use case, all "trajctories" are sequences of tokens which can in principle be embedded into a fixed-dimension feature space. In existing implementations, we are given a dataset of pairwise comparisons over token sequences which are interpreted as feature vectors. In common variants of RLHF, a reward function is learned to maximize the Bradley-Terry (BT) likelihood over a dataset of such comparisons. LCPO is just an alternative to this step, where instead of maximizing likelihood we solve a small number of linear programs. Just as in the case of BT, LCPO can be used on a dataset which contains only a small subset of possible pairwise comparisons. Once the reward function is obtained using LCPO, it can be used for reinforcement learning in precisely the same way as the standard RLHF pipeline.
>
> > I feel that the linear social choice paradigm may be too restrictive. I believe that the authors focus on linear models, because this is indeed a natural machine learning model. However, I don't think it is necessary to assume that the rankings of voters (i.e., humans in AI alignment) have to be linearly representable. In my opinion, the paper would be stronger if the results were presented without this latter assumption.
>
> While the linearity assumption is admittedly a limitation, we note that linear reward models are more general than they may first appear, as the "feature vectors" can be output embeddings of a pretrained LLM (and we just learn a linear layer of a reward model from pairwise comparisons). Consequently, even a linear model can in principle encode highly non-linear characteristics of the raw input (e.g., text).

---

> > ### Comment · Reviewer_TLca · 2024-08-09
> >
> > Thank you for the clarifications.

---

### Comment · Area_Chair_aAHc · 2024-08-08
**Reviewer - Author Discussion**

Thanks everyone for their hard work on the papers, reviews, and rebuttals. We now have a comprehensive rebuttal from the authors which responds both overall and to each review.

I'd please ask the reviewers to please post a comment acknowledging that they have read the response and ask any followup questions (if any). Thanks to those of you who have already done this!

This period is to be a discussion between authors and reviewers (Aug 7 - Aug 13) so please do engage now, early in the window, so there is time for a back and forth.

Thanks!

---

### Decision · Program_Chairs · 2024-09-25

**Decision:**

Accept (spotlight)

**Comment:**

After a good set of reviews, discussion, and rebuttal that involved all reviewers and the author we have arrived a recommendation of accepting this paper for NeurIPS. The paper is well written, within scope for the conference, and provides a solid theoretical contribution in the direction of LLM alignment.